# Diversity of Species and the Occurrence and Development of a Specialized Pest Population—A Review Article

**Anna Wenda-Piesik [1],\* and Dariusz Piesik [2]**

[1]  Department of Agronomy, UTP University of Science and Technology, 7 Kaliskiego, 85-796 Bydgoszcz, Poland
[2]  Department of Biology and Plant Protection, UTP University of Science and Technology, 7 Kaliskiego, 85-796 Bydgoszcz, Poland; piesik@utp.edu.pl
\*  Correspondence: apiesik@utp.edu.pl

**Abstract:** The trophic interactions between plants and herbivorous insects are considered to be one of the primary relationships in the occurrence and development of specialized pest populations. Starting from the role of multicropping and the types of mixtures through the ecological benefits of intercropped plants, we explain the ecological conditions that contribute to the occurrence of pest populations. The dynamics of pest populations in crop occur in stages with the survival and development of pest in source of origin, invasion and distribution in crops, development and survival of the population, emigration to the another crop and (or) change of habitat. Possible effects of each stages are described based on the camouflage of visual effects, olfactory effects and reversal of feeding preferences. Fundamental theories of natural enemies and concentration of food resources have been explained to refer to the empirical data.

**Keywords:** intercropping; mixed crop; herbivores; pest population; natural enemy



## 1. Introduction

Agrosystems provide the food source for the human population which are vulnerable to serious quantitative and qualitative losses due to the occurrence of specialized crop pests [1,2]. Agrocenoses, through their floristic compositions, can regulate the change patterns of diversity and ecological processes between plants and pests through a variety of mechanisms, particularly trophic and behavioral regulation [3–5]. Throughout the world, monoculture (single species) cropping is the most intensive method of plant production. It is the most simplified cultivation method and its aim is to maximize yield and net profit. However, the growth of monoculture system is associated with biological problems: monocultures are more susceptible to pests, diseases and weeds. As monoculture continues, the phytosanitary condition becomes increasingly unstable and requires absolute chemical protection via intensive programs. Pest control in monocultures is based primarily on the use of chemical plant protection products of all generations of pesticides [6,7]. An alternative approach to growing some crop species is inter- and intra-species intercropping. Such crops are subjected to less pest pressure and can therefore be controlled without the intervention of chemical agents [8].

The greater the degree of differentiation in agroecosystems, the more stable the systems that regulate pest populations become when compared with monocultures and productivity is not as compromised [9,10]. A phenomenon that positively influences the efficiency of mixed crops is complementarity [11], due to different species being able to make better use of the habitat's resources, which, in turn, translates into increased plant productivity and total yield [12]. Mixed crops can also better counteract soil erosion and degradation of organic matter, contributing to an increase in the content of organic carbon and nitrogen in soil [13–15].

The trophic interactions between plants and herbivorous insects are considered to be one of the primary relationships that occur in agrocenoses. The presence of pests is

regarded as one of the most important biotic factors that affect consecutively cultivated plants during each growing season. The cultivation of only one plant species (especially in monocultures) results in the development of specialized phytophages and, consequently, leads to a reduction in plant productivity. Depending on plant succession, the development of the pest population may be completely or partially limited.

Plant species utilized in crop systems can either improve or worsen the phytosanitary quality of the site for each plant [16]. According to the theory of crop rotation, strategies for the continuation of growth must include preventing so-called crop rotation diseases or, at a minimum, establishing an environment that is not conducive to the excessive development of pest populations [17]. However, crop rotation is not the sole approach that leads to a reduction in populations of pests, pathogens or weeds. Recognizing the crucial role of pest control, researchers are utilizing other methods in the search for new solutions; for instance, resistance breeding or different methods of plant cultivation [18]. In order to reduce the risk of crop failure, which is influenced by the gradations of specialized pest populations, and, at the same time, to ensure crop yield stability, intercropping should be introduced as often as possible into crop production systems [19].

## 2. Multicropping and Types of Mixtures

According to the literature, multicropping is defined as a practice of consecutively sowing different plants in the same field during a single growing season and many different types of multicropping systems exist. In reference to multicropping systems, Andrews and Kassam [20], Perrin [21] and Willey [22] also include the practice of mixed cropping, which involves planting two or more plant species simultaneously in the same field; these different species coexist either for a limited time or for the whole duration of the growing season. On the other hand, multicropping does not involve the following: single species cultivation (in the same field and for the entire growing season), sowing winter plants in the same growing season subsequent to harvesting spring or winter plants or the cultivation of winter crops in monoculture. It also excludes permanent grassland or sowing perennial monospecies grasses or small-seed legumes on arable land.

Considering the spatial distribution of different plant species and taking into account the length of time that they co-occur, we can distinguish the following types of multi-cropping systems: (1) consecutive crops: during one growing season, two (seldom more) short-term crops, such as mulching crop and spring barley *Hordeum vulgare* L. are sown (in the same field) in successive, relatively short intervals of time; (2) variable crops: a single plant species is introduced into an existing crop of another species. In Poland, this method of cultivation is referred to as "undersowing." The overlap (period of time in which both species coexist) fluctuates from a few to several weeks; for example, seradella *Ornithopus* spp. is introduced into an existing crop of winter rye *Secale cereale* L.; (3) intercrops ("co-crops" or mixed crops): two or more plant species (including varieties of a single species) are cultivated simultaneously. In this instance, the developmental process overlaps in space and time, for example, a mixture of barley and oats *Avena sativa* L.

Intercrop methods can be further divided into the following groups according to their cultivation pattern: (1) rowless plant mixtures: plants of different species (or plant varieties) are sown according to assumed proportions; their placement, however, occurs at random, which results in an unsystematic or mosaic crop pattern, for example, sowing clover seeds with ryegrass *Lolium perenne* L.; (2) row crop mixtures: a mixture of two or more plant species is attained by placing seeds in regular rows but with an irregular distribution pattern within the rows, for example, a mixture of barley and peas *Pisum sativum* L.; (3) inter-row cultivation: plants of each species are alternately arranged and placed in separate, uniform rows, such as single, double or multiple rows. This is a special type of strip cultivation that is used primarily for the cultivation of vegetables. It allows for an independent cultivation technology to be used for individual species; (4) coordinated and rowless mixed cultivation: one species is grown in rows and the field distribution of the remaining species is random; for example, spring barley is sown in rows and alfalfa

*Medicago sativa* L. is distributed randomly. Therefore, a crop mixture can be defined as the process of simultaneously cultivating two or more species or varieties of arable crops in the same field. Species or varieties, described as mixture components, are usually sown and harvested at the same time. In special cases, however, both the sowing of seeds and the collection of individual species may be performed at different times. 'Simultaneous cultivation' refers to cultivation in one ecological niche for a significant period of the growing season [20]. Furthermore, multicropping systems also include mixtures: spatially arranged crops (where plants of each species are sown in separate rows) and crops characterized by an irregular presence of species within each of the rows [22].

In traditional field crops, both Asian and some tropical regions have the highest share of mixed crops [23]. This particularly applies to such mixtures as coconut *Cocos nucifera* L. and pineapple *Ananas comosus* (L.) Merr, corn *Zea mays* L. and potato *Solanum tuberosum* L., corn and sweet potato *Ipomoea batatas*, sorghum *Sorghum bicolor* (L.) Moench and peas and beans *Phaseolus vulgaris* L. and corn. The achievements of genetics and breeding programs of grasses and clovers have led to the adaptation of varieties of these species for mixed crops for lawn and forage use. Much attention has been paid to research into multi-cultivar crops in common wheat, *Triticum aestivum* L. and rice, *Oryza sativa* L. It is well known that such crops result in greater productivity per acreage because individual cultivars use habitat resources, such as water, light and soil components, more efficiently [24,25]. In countries where agriculture is less developed, traditional crops have always been mixed because of the scarcity of arable land (rarely exceeding 1.5 ha) [26] and this practice has reduced the risk of crop failure [27]. In Central Europe, intercropping involves the utilization of plants from the *Poaceae* and *Papilionaceae* families. The following mixtures are used: mixed cereals of various species, mixtures of varieties of one type of grain (most often barley), mixtures of legume species, mixtures of cereals and legumes, mixtures of small-seed legume species with grasses and mixtures of grass species. The latest research shows that in the cropping of maize with common beans or garden nasturtium *Tropaeolum majus*, yields of dry matter were obtained in comparable quantities and qualities to those resulting from the cultivation of maize alone. This study showed that the intercropping of maize in Central Europe with flowering partners can be a suitable alternative to growing maize alone and can increase field biodiversity [28]. Corn and common beans in co-cultivation is one of the most common food crop production practices in small farms in Sub-Saharan Africa (SSA). In Europe, other forms of multicropping involve introducing undersown, small-seed legumes (or grasses) into cereals and mulching crops, while the strip system is predominantly used in the cultivation of field vegetables. In Poland, the possibilities of their utilization are limited primarily by the length of the growing season. Therefore, given climatic conditions, only certain types of multicropping techniques can be used. Many years of research conducted in Poland have shown that cereal mixtures (especially barley with oats) produced higher yields than pure crops of the same varieties, mainly due to an increase in the leaf area index *LAI* and lend equivalent ratio *LER* [29]. Comparing the organic management system with the integrated management system revealed that the average gross margin (less profit) was twice as high in the mixtures grown in the organic system [30]. However, when deciding to make changes in crop selection, one should take into account the consequences of decreasing the use of mixed crops, as this may hinder the implementation of self-sufficiency and land-use efficiency programs [31].

## 3. Benefits of Growing Plants in Mixtures

The biodiversity of farmlands has significantly declined, which can be explained by the intensification of agricultural production [32–34]. In consequence, this decline may reduce the abundance of natural enemies and their effects on pest species [35–38].

It is well known that the intensification of agriculture is one of the main causes of biodiversity loss [39] and also has a negative effect on ecosystems [40,41]. Thus, there is a need for more sustainable agricultural practices [42]. Diversification practices (e.g., intercropping or diverse field margins) were intensively used for many centuries and, to

date, are well accepted as one of the most promising practices to maintain the biodiversity of ecosystems. Moreover, they may increase productivity in widely utilized agricultural systems [43].

Intercropping plays an important role in controlling many pest species and protecting beneficial insects, which are essential for enhancing biodiversity in an agroecosystem [44–47]. Not surprisingly, it is also important to consider the degree to which host plants are resistant to aphids (*Aphis* spp.). In the intercropping system, wheat cultivars that are resistant to cereal pests may reduce cotton aphids, *Aphis gossypii* L. more effectively than an aphid-resistant variety [48].

Many scientific activities have highlighted the effects of plant diversification on pests, pathogens and beneficial organisms in the agricultural landscape. The results of these studies suggest that habitat manipulation (e.g., intercropping) and rotation can considerably improve both disease and pest management [49].

Intensive agriculture has achieved many advances in agroecosystem productivity. Intensive cropping systems prefer specialized plant group (e.g., cereals) and replace diverse plant ecosystems with monoculture. This not only leads to the loss of cultivated plant resources but also reduces the numerous benefits provided by biodiversity within agroecosystems (e.g., biological control) [50].

The benefits of multicropping for plant cultivation include the development of plant species, for example, an increase in nitrogen uptake by cereals that are cultivated in a mixture with legumes [51]; the efficient use of solar energy in mixtures of monocotyledonous and dicotyledonous plants [52,53]; the minimization of self-poisoning in some crops [54,55]; the incidence of "soil fatigue" [56,57]; a significantly more efficient use of water and nutrients [58]; soil profile; the complementary use of space [54,57,59]; the formation of dense soil cover [60]; the limitation of pests and crop diseases [56,61–63]; the growth of certain species in conditions that are unfavorable for other species [59,64,65]; and higher productivity of multispecies communities as compared with monospecies systems. Mixed cultivation fully supports the various arguments presented in favor of this type of cultivation system [66–69].

## 4. Ecological Conditions That Contribute to the Occurrence of Pest Populations

The widespread use of chemical plant protection products has caused a number of negative changes and problems, among which an increase in pest resistance and harm to plant pollinators are key effects. In this regard, EU regulations have reduced the spectrum of allowable pesticides and announced a green deal for Europe, according to which the use of pesticides is to be restricted by 30% within 10 years. The use of alternatives to chemical-based methods for pest control, including the increased emphasis on natural enemies, primarily aims to increase the biodiversity of the biocenosis [70,71].

In some integrated pest management systems, the use of mixed crops is a practice to prevent excessive pests. In Mubi, Adamawa and Nigeria, the intercropping of cowpea *Vigna unguiculata* (L.) Walp and sorghum significantly reduced the aphid population (*Aphis craccivora* Koch) compared with the sole crops of these species [72]. However, Oso and Falade [73] stated that intercropping may support other practices but, on its own, may not necessarily resolve increasing pest populations or reduce the pest burden in all situations. Any cropping system with high pest pressure can be managed relatively early as the predator population increases. The start of vegetation growth is always a critical period (autumn in the case of winter crops and spring in the case of spring plants) when the ratio of predator to pest is the highest. It is during this time that the pest population is most likely to be suppressed by predators [74].

Biodiversity is defined as species richness; namely, the variety and variability of species at all trophic levels of any given biocenosis. In complex biocenoses, determining the diversity of animals on a local scale (of any specific ecosystem) depends on the heterogeneity in space, predation and competition. Predation plays a dominant role in shaping the diversity of organisms. In simple biocenoses, however, it is competition that constitutes

the most significant factor in organism diversification, which intensifies with the occurrence of highly specialized herbivores that exhibit strong preferences for narrow ecological niches [75]. Diversity in any ecosystem should be treated holistically and it is crucial to understand and carefully consider the role of all trophic levels within it. Among strategies that aim to reduce pesticides, biological pest control is the safest and pro-ecological service for the entire natural environment. However, increasing reservoirs of natural enemies and their population sizes has rarely been the subject of research. The dynamics of the populations of pests and their natural enemies require time and the maintenance of ecological control mechanisms in the agricultural system, which should be studied in the growing cycle and repeated over multiple years [76]. Price et al. [60] emphasized the importance of various interactions between the plant, the herbivorous insect and the herbivore's natural enemies. For example, it is not possible to fully understand the relationship between a plant and its pest without careful consideration of the impact caused by the insect's natural enemies. Consequently, the importance of each trophic level cannot be overlooked.

Barren biocenoses are characterized by relatively small numbers of dominant species and the presence of a substantial quantity of individual species (per unit area). This phenomenon applies to both the producer and consumer levels. These simple systems are significantly more susceptible to an increased presence of a single insect species than any other natural ecosystem [77]. Bey-Bienko [78] provides an example of typical changes in the composition of fauna as a result of a natural system's transformation into arable cropland. The author states that in the natural steppe ecosystem, the number of insect species was 312, while the number of organisms per square meter was about 159. The relationship between the diversity of species and the number of individual species per area unit was inverted after field conversion into a monoculture of wheat. As a result, the number of species dropped to 135, while the number of organisms per square meter increased to 341. Repeated cultivation of the same species in large spaces favors the outbreak of pests. The separation of plants in time (i.e., crop rotation) or in space (multiple crops) can potentially reduce herbivorous insects. In agrocenoses, one approach that results in the differentiation of species or structural differentiation of the canopy involves adding a taxonomically foreign plant to the cultivation of another species or the simultaneous cultivation of genetically diversified plants of the same species. Some authors believe that diversified cultivation requires the presence of undesirable plant species, that is, weeds [79]. Diversified or multispecies cultivation systems contribute to the increased stability of the agrocenosis and, as a practical benefit, the reduction in pest populations [21].

Stability is one of the most important, naturally occurring features of biocenoses. Its disruption or change has a negative effect on the abundance of all populations that exist in the biocenoses. As a result of this stability, it is possible to maintain a relatively consistent influence from disruptive factors [75]. It has been established that the greater the species diversification in any given plant community, the greater the efficiency of the entire trophic network, which affects the balance between the populations of herbivorous and predatory insects [80]. Therefore, an increase in diversity leads to an increase in stability due to properly functioning self-regulating mechanisms of biocenoses. Elton (quoted by Krebs) confirms this thesis by stating: "A sudden explosion in pests' population occurs more often in simple biocenoses or on areas transformed by man" [75]. Because of genetic uniformity and the relatively short period of existence of a given field, agrocenoses are characterized by little biotic diversity and, consequently, limited stability. Therefore, many researchers consider the excessive simplification of agroecosystems to be the primary cause of considerable yield losses [81,82]. It has been estimated that the reduction in global food resources due to pest activity amounts to approximately 13% annually [77].

## 5. Diversity of Crop Species and the Occurrence of Pests

Herbivorous insects exhibit selective preferences towards host plants. In natural biocenoses, plant communities consist of numerous and unrelated species. Herbivorous insects, when looking for a niche that suits their preferences, are guided by chemical or

visual stimulators that emanate from plants. Even insects with a fairly broad foraging spectrum demonstrate food preferences and, therefore, inhabit communities with microclimates that are most suitable to their needs and requirements [64,82]. In mixed crops, the spatial dispersion of hosts is the main factor that influences the dynamics of the insect population. Table 1 provides examples of pests that have altered their behaviors or the development of their populations as a consequence of diversity in crop species.

**Table 1.** Examples of crop pests for which changes in the behavior or development of the population have been observed due to intercropping.

| Name of Pest and Family | Host Plant | Type of Intercropping | Changes in the Pest Behavior and Pest Population | | References |
|---|---|---|---|---|---|
| *Acalymma vittata* Chrysomelidae | Cucumber | Inter-row cultivation, cucumber and corn or broccoli in separate rows | (a) (b) (c) | Three times fewer beetles than in pure cucumber crop<br>Reduction in the reproductive rate<br>Decrease in the period of foraging | [20] |
| *Phyllotreta cruciferae* Goeze Chrysomelidae | Broccoli | Inter-row cultivation, broccoli in rows and white clover between rows | (a) (b) | Colonization of broccoli beetle populations is 1.3 times slower than in the pure stand of broccoli<br>Two-fold increase in the migration time of beetles to other crops | [61] |
| *Phyllotreta cruciferae* Goeze Chrysomelidae | Broccoli | Inter-row cultivation, broccoli in rows and vetch and bean between rows | (a) (b) (c) | Decrease in the foraging period<br>Abandonment of mixed crops<br>Decreasing of the population | [83] |
| *Phyllotreta cruciferae* Goeze Chrysomelidae | Cabbage | Row-crop mixture, cabbage and tobacco or tomato in separate rows | (a) (b) | Significant reduction in the pest population of coordinate-mixed cultivation with shortening of beetle feeding time<br>More than 3 times fewer second-generation beetles as compared with the cultivation in the pure stand | [82] |
| *Aphis craccivora* Koch Aphididae | Groundnut | Row-crop mixture, groundnut and common beans in separate rows | (a) (b) | Common bean's sticky tendrils kept aphids away<br>A reduction in aphids as vectors resulted in a decrease in the virus that causes rosette disease of groundnut | [84] |
| *Oulema* spp., Chrysomelidae | Oat, barley | Row-crop mixture of both cereals | (a) | Mixed cultivation of each species reduces the degree of damage to oat leaves by 48% and barley by 51% compared with pure stands | [85] |
| *Rhopalosiphum padi* L., *Sitobion avenae* L. Aphididae | Barley | Row-crop mixture of barley with yellow lupine and pea | (a) | The number of aphids on barley heads was 3–6 times lower in crops with legumes | [86] |

Perrin lists four aspects that determine the development of the pest population in mixed cultivation: the infestation of the crop by the pest (colonization), the development of its population, the dispersion of herbivores in the cultivation and the presence of natural pests [20]. The individual stages of development of the pest population and the possible effects of changes in insects are presented in Figure 1.

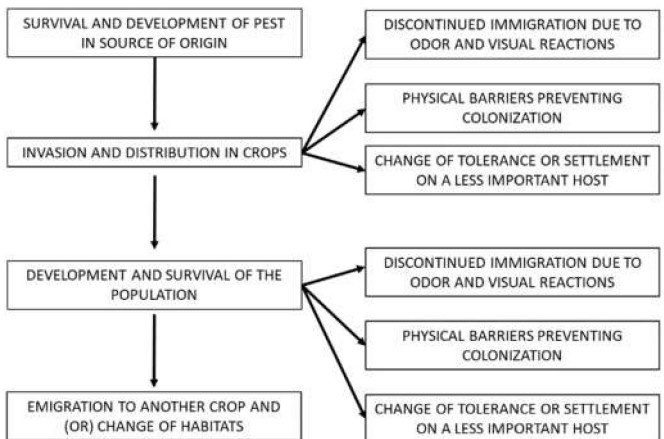

**Figure 1.** Stages in the dynamics of pest populations resulting from mixed cultivation. Possible effects are listed on the right [87].

## 6. Colonization of Crops by Pests

The following factors influence the colonization of mixed crops by specialized pests:

(1)  Camouflage of visual effects. A mixed crop becomes visually unattractive to incoming pests since host plants are often obscured by non-host plants with longer shoots or branch shapes. Consequently, insects' perception of the entire cultivation area becomes skewed. "Foreign" plants constitute a physical barrier to the spread of pests; they also function as "traps" [83,87].

(2)  Olfactory (aromatic) effects. Attractants or feeding stimulants secreted by the host plants play a significant role and determine the way herbivores orient themselves in the environment. Strongly aromatic plants such as tomato *Solanum lycopersicum* L., garlic *Allium sativum* L., onion *Allium cepa* L. and tobacco *Nicotiana tabacum* L. (while cultivated together with other species) may disturb the olfactory perception of the habitat [82,84,88].

(3)  Reversal of feeding preferences. In some cases, pests show strong preferences for and inhabit only certain plant species cultivated in a particular mixture. As a result, pests become "distracted," which, in turn, ensures the protection of other, more valuable plant species. Utilization of this phenomenon is exemplified by the planting of alfalfa on the perimeter of cotton *Gossypium* spp. L. crops in California. Cotton bugs *Lygus Hesperus* L. cause significant damage to cotton plantations. However, their adverse effects on cotton fields are reduced considerably due to the insects' feeding preferences and their apparent attraction to alfalfa [21]. Without the introduction of alfalfa, the insects' impact would be significantly more pronounced.

The above-mentioned factors for reducing the insect colonization of mixed crops are of particular importance in the case of populations of mobile pests that inhabit crops at the beginning of each growing season (e.g., winter beetles looking for complementary feeding crops).

## 7. Development of the Pest Population in Mixtures

A sudden increase in the population of pests occurs when individual species are easily capable of locating food, shelter and favorable conditions for reproduction [89]. Because mixtures reduce the population of host plants, they change the design and physiognomy of the cultivation and have a negative effect on the microclimate of specialized pest species. Farell observed that the sticky leaf tendrils of the common bean were capable of "catching" the aphid *Aphis craccivora* Koch a vector of the peanut virus (peanut mottle virus, peanut stripe virus and peanut stunt virus), thus effectively limiting the development of the insect population [84]. The benefits of mixed cultivation are contingent upon the time of insect emergence in relation to the stage of the plant developmental process. The negative effects

of pests are more pronounced during the most critical stages of plant development since plants are most susceptible to damage during their emergence, as well as during the flowering process [21].

## 8. Pest Distribution in Cultivation

Inhibiting the spread of the pest population is possible when host and non-host plants grow together in a particularly unfavorable system for herbivores. The scattering of the cabbage flea *Phyllotreta cruciferae* Goez9e was inhibited on cabbage *Brassica oleracea* L. var. *capitata* L. when cabbage was grown in a row around the perimeter of a meadow to a much greater extent than in a plot in the same meadow consisting of several rows that were only 45 cm apart [90]. It was found that the same pest in a mixture of broccoli *Brassica oleracea* L. var. *italica* Plenck and vetch *Vicia sativa* L. or field bean *Vicia faba* L. var. *minor* Peterm. wasted considerable time and energy on disentangling from vetch shoots and finding the right host among horse bean plants, which resulted in a rapid reduction in its population [83]. The decreased availability of the niche and large distances between plants reduce the relative quality of the insect environment, which, in turn, may lead to the emigration of pests to other, more attractive crops. This effect, in which one plant helps another plant to defend itself effectively against pests, is called "companion immunity." Examples of crops with this type of resistance are listed in Table 1.

## 9. The Role of Natural Enemies

The relationship between a plant and an insect cannot be considered without taking into account the third trophic level: natural enemies, which are considered plant allies [77]. The more diversified the cultivation, the greater the variety and abundance of herbivorous predators and parasitoids. Therefore, the simultaneous cultivation of several species may alleviate and/or stabilize the relationship between a pest and its natural enemy [91]. Long-term crops are of particular importance here, since the stability of the relationship between plant, phytophage and entomophage is positively influenced by an extended period of time [21].

Grape phylloxera, *Viteus vitifoliae* (Fitch), is regarded as the most economically important pest worldwide for commercial grapevines (*Vitis* spp.). Grape phylloxera causes the most economic damage in its root-feeding stages as compared with leaf-feeding stages [92]. Research on grape phylloxera has been extensive because this pest ravaged European vineyards and most of the basic work on phylloxera biology and control was carried out prior to 1920. Granett et al. [93] summarized the major constraints that explain why chemical control has been inefficient in root-galling grape. No efficient biological control method has been developed to date, though many general natural enemies of phylloxera exist [94]. An organic management strategy could reduce root necrosis but it produces no effect on the number of phylloxerae: this observation may be due to the microbial ecology and soil suppression of pathogens [95]. Soil type may also influence phylloxera survival and its spread [96]. However, control methods that may be efficient and practical for supporting successful pest control remain unclear and require more testing.

The effects of grape–tobacco intercropping on populations of grape phylloxera were evaluated in a field in which egg and nymph mortality and female fecundity were significantly affected. It was reported that grape phylloxera populations in the intercropping systems were lower compared with the monoculture pattern and they decreased each year. Vine trees were in better condition upon continuous intercropping with tobacco [97]. Intercropping is also effective in reducing mantis cruciferous *Plutella xylostella* L. populations but the underlying mechanisms are elusive [98]. For example, when exposed to three different types of host plants (*Brassica campestris* L., *B. juncea* Coss. and *B. oleracea* L.), the flight frequency of adult *P. xylostella* females increases, while its fecundity is weakened [99]. Many researchers have studied the positive or negative impact of the infestation of pest species [47,100,101]. Bregante and Matta [102] studied the intercropping of corn and bean and Omar et al. [103] conducted field trials to study the effect of intercropping cotton and

cowpea on the populations of aphids, whitefly and bollworm. Ma et al. [104] examined the strip cropping of wheat and alfalfa to improve the biological control of the cereal aphid, *Sitobion avenae* (Fabr.) by the mite, *Allothrombium* Berlese (Acari: Trombidiidae). It is well documented that wheat–garlic intercropping can reduce the population of *S. avenae* by promoting natural enemies [105]. Similar studies have also been performed in wheat and oilseed rape, *Brassica napus* L. [106], cowpea and sorghum [72] and wheat and pea [107,108].

## 10. Theories of Natural Enemies and Concentration of Food Resources

During the examination of pure as well as mixed cabbage crops, Root [90] observed that the number of pests and their average biomass per 100 g of consumed food was always higher in pure cabbage sowing. In order to explain this phenomenon, the author presented two hypotheses. The natural enemies hypothesis attributes the lower pest density to a more diversified environment, where higher numbers of predator species and insect parasitoids are present and the abundance of their populations is increased [77,109]. Proponents of this theory regard the enemies of natural insects as the main factor in regulating populations of pests. An alternative hypothesis is derived from the theory of food resource concentration. In non-uniform, short-term crops, the effectiveness of natural enemies in reducing phytophages may not be as effective as the mere fact of decreased food concentration. In mixed sowing, specialized pests are deprived of a sufficient food supply, proper breeding base and adequate shelter. Therefore, they show a distinct preference towards single-species compact sowing, where the concentration of host plants is sufficient to maintain all necessary vital functions [90]. Most researchers strongly support the hypothesis of the concentration of food resources [61,79,81–83,87].

On the other hand, opponents argue that the two general theories that explain the interaction between an insect and cultivation in a multiple-plant system cannot be applied to individual pests and their populations. Speaking against the hypothesis of resource concentration, Helenius [110] gives the example of the cereal aphid, *Rhopalosiphum padi* L. in a mixture of oats and field beans. The substantial abundance of the oat plants resulted in a greater density of aphids due to the more pronounced aggregation of colonies established by re-emigrants on a single plant. The activity of natural enemies may also decrease in crops with a variety of species, especially if they become attracted by specific visual or olfactory stimulants, the reception of which may be disturbed by "concealment" by other plant species. Smith [111] postulates that this mechanism causes a disruption in the proper perception of the habitat by the infiltrating herbivorous insects. Moreover, increasing crop biodiversity, such as by strip intercropping, can promote biological pest control in agroecosystems [74,112].

## 11. Conclusions

The spatial and/or temporal separation of host plants is contingent upon the behavior and development of herbivorous insects. Reducing pest populations can be realized by recognizing and identifying their feeding preferences. The more pronounced the feeding preferences, the greater the reduction in the population. Consequently, the damage to host plants grown in mixed sowing systems will be considerably reduced. Monophagous insects are specific in this regard. The slight alteration of a host plant's canopy renders monophagous insects unable to locate an adequate food supply and to establish a suitable breeding base. A significant reduction in the population of oligophagous insects (insects whose host spectrum is in the botanical family) is expected to occur in mixtures of adequately spaced botanical taxa, for example, damage to cereal plants can be reduced by introducing the cereal leaf beetle to cereal–legume mixtures. Numerous empirical data and some theoretical considerations suggest that, in mixed crop systems, the reduction in pest populations is predominantly linked to the availability of food sources and less so to an impact or threat posed by their natural enemies.

**Author Contributions:** Conceptualization, A.W.-P. and D.P.; formal analysis, A.W.-P. and D.P.; investigation, A.W.-P.; resources, D.P.; writing—original draft preparation, A.W.-P.; A.W.-P. and D.P. All authors have read and agreed to the published version of the manuscript.

**Funding:** This research received no external funding.

**Institutional Review Board Statement:** Not applicable.

**Informed Consent Statement:** Not applicable.

**Conflicts of Interest:** The authors declare no conflict of interest.

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
