# Peer review of "Diversity of Species and the Occurrence and Development of a Specialized Pest Population—A Review Article"

_agriculture, doi:10.3390/agriculture11010016_

Round 1
Reviewer 1 Report
A well written paper with suggested changes noted on paper itself.

Author Response
We are grateful to respond to the comments of the reviewers. Thanks to these comments, the manuscript improved in quality.
Regarding comments from the first reviewer.
We highlighted the changes. All comments that were marked in the body text by the first reviewer have been taken into account. All described species of plants and animals have been supplemented by their latin names.
Reviewer 2 Report
The article is interesting for plant protection specialists, contains new thoughts, views on the problem of pest control.
At the same time, there are technical flaws that prevent the publication of the manuscript, for example:
- The names of the authors who described the species of both plants and animals should be given in full, and not abbreviated.
- Authors indicate authors' names for plants, but not for animals. For example, in one case, the authors write Linn (line 339), and in the other L. (line 338).
- Identical generic names should be shortened, for example line 329
- It is necessary to check the relevance of the Latin names of all zoological objects in the fauna-eu.org database. For example errors were found:
line 311 - Daktulosphaira vitifoliae (Fitch) - Viteus vitifoliae (Fitch, 1855)
https://fauna-eu.org/cdm_dataportal/taxon/81445ccd-412b-474c-a64a-7b6eebe8f9f2
line 335 - Allothrombium ovatum
https://fauna-eu.org/cdm_dataportal/taxon/eb50f121-5b13-48f7-8a0e-211edd8ba982
- The names of insect families do not need to be italicized, for example, in Table 1.
- The numbering of the sections of the article and its absence is not clear, for example, line 280, 291, 303.
- In the bibliography, the honor of the names of the journals is abbreviated, and some are given in full. Journal abbreviations do not follow the rules, for example line 529, 534.
- Source 67, 68, 69 and others are not designed according to the rules.
Author Response
Cover letter
We are grateful to respond to the comments of the reviewers. Thanks to these comments, the manuscript improved in quality.
Reviewer the 2nd
- The names of the authors who described the species of both plants and animals should be given in full, and not abbreviated.
Res. According to the rules given in Agriculture edition we used the abbreviations in latin names.
- Authors indicate authors' names for plants, but not for animals. For example, in one case, the authors write Linn (line 339), and in the other L. (line 338).
Res. Uniformly all the abbreviations have been used.
- Identical generic names should be shortened, for example line 329
Res. Corrected – Line 306
- It is necessary to check the relevance of the Latin names of all zoological objects in the fauna-eu.org database. For example errors were found:
line 311 - Daktulosphaira vitifoliae (Fitch) - Viteus vitifoliae (Fitch, 1855)
Corrected – Line 289
https://fauna-eu.org/cdm_dataportal/taxon/81445ccd-412b-474c-a64a-7b6eebe8f9f2
line 335 - Allothrombium ovatum
https://fauna-eu.org/cdm_dataportal/taxon/eb50f121-5b13-48f7-8a0e-211edd8ba982
Res. Corrected – Lines 311-312
- The names of insect families do not need to be italicized, for example, in Table 1.
Res. Corrected – Table 1
- The numbering of the sections of the article and its absence is not clear, for example, line 280, 291, 303.
Res. The numbering has been completed – Lines 258, 269, 282.
- In the bibliography, the honor of the names of the journals is abbreviated, and some are given in full. Journal abbreviations do not follow the rules, for example line 529, 534.
- Source 67, 68, 69 and others are not designed according to the rules.
Res. Regarding the references – all was checked and abbreviated